# Incorporation of the Dry Blossom Flour of *Sambucus nigra* L. in the Production of Sponge Cakes

**DOI:** 10.3390/molecules27031124

**Published:** 2022-02-08

**Authors:** Galia Gentscheva, Iliana Milkova-Tomova, Dragomira Buhalova, Ivaylo Pehlivanov, Stefan Stefanov, Krastena Nikolova, Velichka Andonova, Natalina Panova, Georgi Gavrailov, Tsanka Dikova, Zhivka Goranova

**Affiliations:** 1Department of Chemistry and Biochemistry, Medical University-Pleven, 5800 Pleven, Bulgaria; 2Department of Nutrition and Tourism, University of Food Technologies, 4020 Plovdiv, Bulgaria; iliana_tomova@abv.bg (I.M.-T.); dra.buhalova@gmail.com (D.B.); 3Department of Pharmaceutical Technologies, Medical University-Varna, 9000 Varna, Bulgaria; Ivaylo.Pehlivanov@mu-varna.bg (I.P.); stefan.stefanov@mu-varna.bg (S.S.); Velichka.Andonova@mu-varna.bg (V.A.); Georgi.Gavrailov@mu-varna.bg (G.G.); 4Department of Physics and Biophysics, Medical University-Varna, 9000 Varna, Bulgaria; nkpanova@gmail.com; 5Department of Dental Materials and Propaedeutics of Prosthetic Dentistry, Medical University-Varna, 9000 Varna, Bulgaria; tsanka_dikova@abv.bg; 6Department of Food Technologies, Institute of Food Preservation and Quality-Plovdiv, Agricultural Academy, 4000 Plovdiv, Bulgaria; jivka_goranova@abv.bg

**Keywords:** texture parameters, quality properties, color characteristics, sponge cakes, *Sambucus nigra* L., functional foods

## Abstract

The present study aims to develop recipe compositions and technology for producing sponge cakes from wholemeal flour, partially replaced with a functional plant component dry blossom flour of *Sambucus nigra* L. Three designs of sponge cakes with 5, 10, and 15% content of flour of *Sambucus nigra* L. corrected up to 100% with whole-grain oat flour were studied. Their characteristics were compared with sponge cakes of 100% wheat flour/control. The obtained new products were characterized by reduced carbohydrates, increased content of dietary fiber, and preserved volume compared to the control. The physicochemical parameters of sponge cake and marshmallows with different concentrations of dry flowers of *Sambucus nigra* L. included in them differed from the control with lower water absorption, pH, and moisture, while having a higher relative mass and ash content and retaining the original size. Pathogenic microorganisms such as *Escherichia coli*, *Salmonella* sp., and *Staphylococcus aureus*, and common coliforms were not detected in the control and experimental samples when determining the microbiological parameters. Therefore, the developed formulations are an excellent alternative to wheat flour, significantly improving some nutritional characteristics such as smell, taste, dietary fiber, and lower carbohydrate content.

## 1. Introduction

*Sambucus nigra* L. is one of the most common plant species in Europe [1,2,3,4]. Its blossoms contain bioactive flavonoids (0.7–3.5%), phenolic acids (5-caffeoylquinic acid and 1,5-dicaffeoylquinic acid), rutin—21.0 mg/g dry substance, and others [5]. Sambucianin, which is contained in the blossoms of *Sambucus nigra* L., along with anthocyanins, glucosides, and flavonoids, protects the cell membranes from the changes caused by the free radicals [6]. These compounds possess antimicrobial, antiviral, anti-inflammatory, diuretic, and spasmolytic properties, and they strengthen the blood vessels [7]. The etheric oil in the blossoms of *Sambucus nigra* L. have perspiring, expectorating, anti-inflammatory [8,9,10], and cardioprotective effects [11], and they suppress the growth of cancer cells [12].

Some of the modern tendencies in nutrition include decreasing the intake of lipids, change in the traditional technologies, and creation of new prescriptions with optimal biological and energetic values in the production of pastry. Sponge cakes are the most commonly used in the making of bakery products, as they have great porosity and can take up to 60% of sugar syrup with different concentrations without losing their elasticity. Their humidity is about 36–38%, and the baking temperature is around 170–240 °C for the time of 10–75 min, depending on the thickness of the cake’s layers [13,14]. The main quality indicators of the semi-manufactured sponge cakes include: (1) 10–30% increase in the initial volume; (2) smooth and thin crust with a golden brown to a dark brown hue [15,16,17,18]; and (3) soft in the middle, with an elastic structure and thin-walled pores identical in size. All of these qualities of the sponge cakes guide the efforts of the sweets manufacturers to produce functional products based on them [19].

The inclusion of plant parts as a food supplement benefits the antioxidant and anti-inflammatory defenses of the organism [20] and provides natural prevention against the development of inflammation and oxidative stress, including obesity, insulin resistance, metabolic syndrome, and type 2 diabetes [21]. The popularization of the idea of a healthy lifestyle leads to avoiding the usage of synthetic antioxidants in different food products and increases the interest in plant-based, natural, non-toxic, and healthy antioxidants. The aim of this research is the development of a prescription for pastry and its production technology, which is based on sponge cake made of wholegrain flour, partially replaced by a functional plant component—dry blossom flour of *Sambucus nigra* L./elderberry/and the evaluation of their physical and chemical properties, as well as their sensory properties.

## 2. Results

Table 1 presents the results of the investigations of the chemical parameters of the different types of flours used in the study since they largely determine the technological properties of the products.

Physicochemical parameters of pandispan blends were determined according to the formulations and technologies developed with *Sambucus nigra* L. dry blossom flour (5, 10 and 15%). The results are presented in Table 2.

Figure 1 and Figure 2 present the main color indications of the crust and crumb cake for the samples studied. The color difference between samples and control is presented on Figure 3.

A summary assessment of the sensory analysis performed by all the tasters is presented in Figure 4.

The textural characteristics of the samples are presented in Table 3. Linear correlations (Table 4) were found between the hardness and elasticity of the sponge cake, as well as between the hardness and concentration of *Sambucus nigra* L. flour.

The photo of the investigated samples is presented in the Figure 5. The distribution of their pores, by microscopic imaging, is shown in Figure 6.

The results obtained from the chemical composition and energy value of the sponge cake layers with *Sambucus nigra* L. blossom flour are presented in Table 5 for a 100 g product.

## 3. Discussion

The analyzed wholegrain oat flour and dried blossom flour of *Sambucus nigra* L. are good sources of biocompounds, and are therefore suitable in the development of functional foods. *Sambucus nigra* L. dry blossom flour has a low protein content and no fat. The presence of ascorbic acid confirms that the blossom of *Sambucus nigra* L. has functional health potential. Wholemeal oat flour has a higher protein, fat, dietary fiber, and mineral content than wheat flour. The fat content of wholegrain oat flour is the highest.

The addition of *Sambucus nigra* L. blossom flour and oats resulted in an increase in the relative mass from 0.72 ± 0.03 to 0.81 ± 0.01 of the new product.

The volume of the sponge cake layer control (207 ± 7 cm^3^) was similar in value to that of the sponge cake with the addition of elderberry flour.

The control has the highest moisture percentage, and the ash content is inversely related to moisture. The water absorption capacity of the elderberry flour products was lower compared to the control (333 ± 8 g) and decreased with increasing concentration of the functional component. The addition of *Sambucus nigra* L. flour (5, 10, and 15%) lowered the pH of the product compared to the control. This is probably caused by the lower pH of *Sambucus nigra* L. flour (5.32) than pH of control (5.58). The blends with functional components contained more minerals, leading to an increase of the ash content, with the blend with 15% elderberry flour having 2.98 times more ash content than the control. Weight losses after baking are calculated according to Hathorn et al. [22]. For control they are 10.2%, while for the samples with *Sambucus nigra* L. weight losses are up to 11.7%.

Moisture content, textural parameters (hardness, elasticity, and brittleness), fat content, carbohydrate content, color parameters a* and b*, lightness (L), and color difference ∆E are important characteristics in the quality control of the sponge cake layers.

When controlling the quality of sponge cake layers, one of the main criteria is lightness. For baked and fried products, it is known that dark brown color is not preferred by consumers [23]. The use of 5% and 10% dried elderberry blossom flour resulted in the preservation of the lightness of the crust and middle part as that of the control. Maximum values for the indicator “L” for crust and middle part were recorded in the sponge cake layer with 15% *Sambucus nigra* L. Therefore, 15% dried elderberry blossom flour had the greatest darkening compared to the control. High values of b (yellow component) and values of a > 2 (red component) indicate a significantly bright and intense orange–brown color [24]. The color difference of the bark between the control and the tested samples shows that in the products containing *Sambucus nigra* L. it remains practically the same. There was an increase in color difference only for the center of the sponge cake blotch with 15% dry flower meal of *Sambucus nigra* L. (Figure 3). Therefore, the addition of *Sambucus nigra* L. flower flour above 10% can lead to effective color change during roasting. Similar results in the literature have been reported for chia oil powder supplementation [25]. A good correlation was found between bark lightness and moisture content in the pandishpan bogs. It is known from the technology that the Maillard reaction and caramelization processes are significantly dependent on moisture content; a similar finding was reported by [26] for fried sweet potatoes and by [27] for products fried by Air Frying technology.

The saturation ‘‘C’’ of a color is a metric defining the visual perception of color, by which the quality of the color is characterized. All sponge cakes had a color saturation above that of the control. The hue “h “can be represented quantitatively. Different shades were discovered in the center of the sponge cake layers containing *Sambucus nigra* L. blossom flour, which clearly correlated with the brightness and saturation indications.

For the indications ‘color’, ‘smell’, and ‘sweet taste’, the sensory evaluators gave the highest scores for the sample with 15% *Sambucus nigra* L. (Figure 4). The values for the indicator ‘shape’ of the sponge cakes were high. Therefore, the addition of *Sambucus nigra* L. dried blossom flour does not adversely affect the shape of the newly developed products. That is also confirmed by the physicochemical data of sponge cake layers (Table 2) for the layer volume parameter. Sponge cake layers with concentrations of 10% and 15% *Sambucus nigra* L. flour in the tested samples had the highest values for the “shape” parameter, regardless of the added fibers. Tasters gave the highest marks for the size and uniformity of the sponge cakes with 5% and 10% dried *Sambucus nigra* L. blossom flour. With the greatest middle part softness, equal in value to the control, was the sponge cake layer with 10% elderberry blossom flour.

The smell of the sponge cakes containing *Sambucus nigra* L. blossoms in the form of flour is not typical for bakery products, but is perceived by sensory evaluators as more pleasant compared to the smell of the control sponge cakes.

The intensity of the sweet taste of the *Sambucus nigra* L. flower marshmallows increased in direct proportion to the increasing amount of *Sambucus nigra* L.; it was also reported that at high percentages of the additive (10% and 15%), albeit faint residual aftertaste was detectable. The control sponge cake and the one with 5% added *Sambucus nigra* L. blossom flour had the lowest residual aftertaste values.

As the concentration of *Sambucus nigra* L. dry blossom flour increased, the hardness of the samples increased linearly relative to that of the control. The increase in hardness is probably associated with lower moisture content; similar information has been reported by [28]. Elasticity characterizes the ability of a sample to recover its shape after the application of an appropriate pressure by measuring the rate of return of the sponge cake to its original state after the force that caused the deformation has been removed [29]. The results obtained are less than 10% different. Linear correlations (Table 4) were found between the hardness and elasticity of the sponge cake, as well as between the hardness and flour concentration of *Sambucus nigra* L.

As can be seen from the results obtained, increasing the concentration of *Sambucus nigra* L. flour did not affect the pore size, but increased the hardness (15% flour) by two to three times compared to the control, and the tenderness increased.

The high percentage of dietary fibers in the product with 15% *Sambucus nigra* L. blossom flour indicates that it has functional potential and can be labeled ‘high in dietary fibers’ [30]. This high dietary fibers content allows a health claim to be made with a contribution to normal bowel function [31], in addition to a reduction in the percentage of carbohydrates compared to the control by about 7% (Table 5).

The microbiological parameters of the samples were investigated, since the plant raw materials can be contaminated with spore-forming bacteria, including the human pathogen *Bacillus cereus*; regardless of the heat treatment, the spores can survive and subsequently overgrow and cause spoilage of the finished products. The toxins formed to make the food dangerous to human health. The intake of such foods causes nutritional diseases. Therefore, the experimental samples with the addition of *Sambucus nigra* L. flour were tested for the presence of *Bacillus cereus*. In the determination of the microbiological indicator molds and yeasts, the presence of *Bacillus cereus* was not detected, but only the presence of mold fungi in the control and the samples with 10 and 15% dried *Sambucus nigra* L. blossom flour, with the number of spores of microscopic fungi ranging from 10 to 20 cfu/g. The results obtained indicate that the prepared samples and the control meet the requirements of the standard, according to which the amount of spores of microscopic mold fungi is up to 100 cfu/g. For the 5% sample, no mold fungi or yeasts were detected.

Regression linear models were established, taking into account the content of elderflower flour on the qualities of the cupcakes at a significance level of 0.01 (Table 4).

## 4. Materials and Methods

### 4.1. Materials

In order to produce sponge cake dough, the following materials were provided by some of the big retail chains in Bulgaria: wheat flour “Extra” (type 500)—manufactured by “Sofia MEL” JSC, Sofia, hen eggs, sugar (crystal, refined); and wholegrain oat flour—manufactured by “Ecosem Bulgaria” Ltd., Stambolovo village.

The blossoms of *Sambucus nigra* L. were collected during the period of intense flowering (May–July), depending on the altitude of the region [32]. The identification of the plant material was carried out at the Department of Plant and Fungal Diversity and Resources, Institute of Biodiversity and Ecosystem Research at the Bulgarian Academy of Sciences (IBEI-BAS), Sofia, Bulgaria. The specimen identification number is SOM 1406.

The blossoms of *Sambucus nigra* L. are from locations in the Western Rhodopes: town of Velingrad, Golyam Beglik Dam, Karatepe-Sutka locality (Bulgaria) with geographical coordinates 35TKG5111032486, Lat. 41. 80444, Lon. 24. 159444, located at 1271 m above sea level. Blossoms of *Sambucus nigra* L. were dried in a thin layer, in the shade, being turned periodically to a constant mass of the sample. Dry samples were ground to a fraction size of 1–2 mm.

Recipe compositions were developed—(1) control–sponge cake; (2) sponge cake blends with partial replacement of wheat flour with 5%, 10%, and 15% *Sambucus nigra* L. blossom flour and oat flour. The exact recipe compositions of the control and the newly developed blends are given in Table 6.

In further interpretations we will use the following designations: Option 1—with 5% elderberry flour + 25% wholemeal oat flour; Option 2—with 10% elderberry flour + 50% wholemeal oat flour; Option 3—with 15% elderberry flour + 85% wholemeal oat.

### 4.2. Methods for the Study of Sponge Cake Layers

The active acidity of the dough and the middle part of the cake layer were according to AACC 02-52 (27), pH meter (model HANNA pH 211, Laval, QC, Canada). The temperature of the dough—(20 ± 0.5) °C.

Moisture content (total) was determined by standards for regulation for taking samples, determination of dry material, moisture, ash and ash alkalinity [33,34]. Each total moisture value is an average of three measurements; volume was measured according to [35]. 

The relative mass of the sponge dough was calculated according to AACC method 10–95 [35].

The total ash content for the sponge cake layer was determined by firing in a muffle furnace (LM 312. 11, Berlin, Germany) at 700 °C to constant mass [34]. Determination of the amount of fat was carried out by the Soxhlet method by extraction with petroleum ether according to [36].

The amount of total carbohydrates was determined spectrophotometrically by the phenol-sulfuric acid method measuring absorbance at 490 nm [37]. Determination of total dietary fibers was made according to [38].

### 4.3. Determination of the Energy Value

The energy value was determined per 100 g of product in accordance with Regulation No. 23 of the Ministry of Health of 2001 on the conditions and requirements for the presentation of nutritional information in the labeling of foodstuffs. The energy value of the marshes was calculated by applying d’Atwater’s energy coefficients to the major nutrients. Accordingly, for 1 g protein = 17 kJ (4 cal), 1 g digestible carbohydrate (without dietary fiber) = 17 kJ (4 kcal), 1 g fat = 37 kJ (9 kcal), and 1 g dietary fiber = 8 kJ (2 kcal).

Figure 7 and Figure 8 show the process flow charts for the preparation of the sponge cake control and those with 5%, 10%, and 15% blossom flour of *Sambucus nigra* L. 

### 4.4. Methods for Microbiological Analysis of Sponge Cake Layers

Microbiology is made according to [39,40,41,42,43].

### 4.5. Color Characteristics

Measurement of the color of the peel and the sponge cake middle parts were made using a quality technical colorimeter PCE-CSM 5, PCE Instruments UK Ltd (Southampton, UK). The indicators are reported by the CIELab system [44]. In the measurement, the color parameters a and b and the luminance L were determined.

The color tone and the color tone angle were determined according to dependencies, respectively:
(1)
$$C_{\mathrm{ab}}=\sqrt{a^{2}+b^{2}}$$


(2)
$$h_{\mathrm{ab}}=\mathrm{arctg}\left(b/a\right)$$


The total color difference (ΔE) was calculated from Equation (3), where *L*_0_, *a*_0_, and *b*_0_ are the color parameters of the control sample.

(3)
$$\Delta E=\sqrt{{\left(a-a_{0}\right)}^{2}+{\left(b-b_{0}\right)}^{2}+{\left(L-L_{0}\right)}^{2}}$$


All measurements were taken at room temperature with the experiment repeated five times.

### 4.6. Texture Profile Analysis

The textural properties of sponge layer cakes with 5, 10, and 15% dry blossoms of *Sambucus nigra* L. were measured with a texture analyzer (Belle texture analyzer, Agrosta Overseas, Serqueux, France) using special software. A Teflon press (40 mm diameter) was used to obtain textural parameters such as hardness, elasticity, and brittleness under double compression. The test was carried out on pieces of pandishpan in the shape of a truncated cone. 

The parameters of the analysis are as follows: Pre-test speed 5 mm/s; test speed 0.25 mm/s; and 10-s delay between the first and second compression. The measurements were performed three times each, and the average values are presented as a result.

### 4.7. Microscopy of Sponge Cake Layers

The middle part of the baked sponge cake layers was observed with an Olympus SZ 61 stereomicroscope, Olympus ltd., Tokio, Japan, operating in the bright field with reflected light at 70× objective and 8× eyepiece magnification. The photos were taken with a 5Mp× Wi-fi camera.

### 4.8. Sensory Evaluation

A descriptive sensory profiling test [45] was used to determine the parameters—shape, color, odour, pore size and uniformity, sweet taste, and residual taste (aftertaste) [46] of newly developed sponge cake blends. To fully describe the characteristics of the products, indicators and maps were prepared to quantify their intensity. Intensity scores for each trait were analyzed statistically to determine significant differences between the products evaluated. The accuracy of the assessment is guaranteed as the cards were completed by a select committee of 10 trained sensory assessors. The marshmallow samples were prepared 1 h before evaluation. The middle swales of each species were cut to 150 × 150 × 150 mm and stored in coded containers covered with aluminum foil. A 10-point linear scale reflects the intensity of each sensory indicator. Coded samples were presented simultaneously and scored in random order. 

### 4.9. Statistical Processing

All parameters studied were determined after three to five repetitions. Data from all studies were processed to obtain the mean and standard deviation (SD). The linear dependencies between the studied parameters were obtained by the least squares method. The coefficient of determination R^2^ was determined.

IBM SPSS software was used in the statistical analyses at the significance level *p* < 0.01.

Kendall’s method—the sensory analysis data of pandichpan blots were processed using Kendall’s concordance method (Kendall’s concordance) to check the agreement in the rankings of the trained tasters in the sensory analysis with respect to the samples studied [47].

## 5. Conclusions

The physicochemical parameters of sponge cake and marshmallows with different concentrations of dry flowers of *Sambucus nigra* L. included in them differ from the control with lower water absorption, pH, and moisture, while having a higher relative mass and ash content and retaining the original size.

Chemical composition—minimal reduction of the amount of protein and carbohydrate content decreases by up to 10%, up to 1.5 times increase dietary fibers, and slightly increases fat.

It also preserves the excellent appearance of the products. The use of *Sambucus nigra* L. flour does not change the microbiological parameters and taste of the products.

Therefore, the developed formulations are an excellent alternative to wheat flour, significantly improving some nutritional characteristics such as smell, taste, dietary fiber, and lower carbohydrate content.

## Figures and Tables

**Figure 1 molecules-27-01124-f001:**
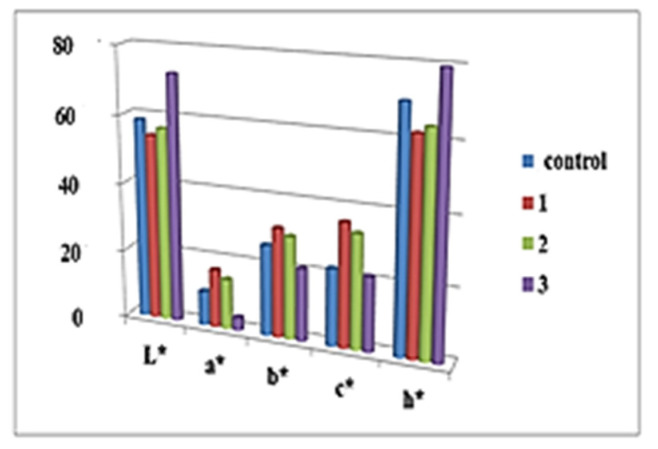
Color indicators of sponge cake crust with elderberry blossom flour (*Sambucus nigra* L.).

**Figure 2 molecules-27-01124-f002:**
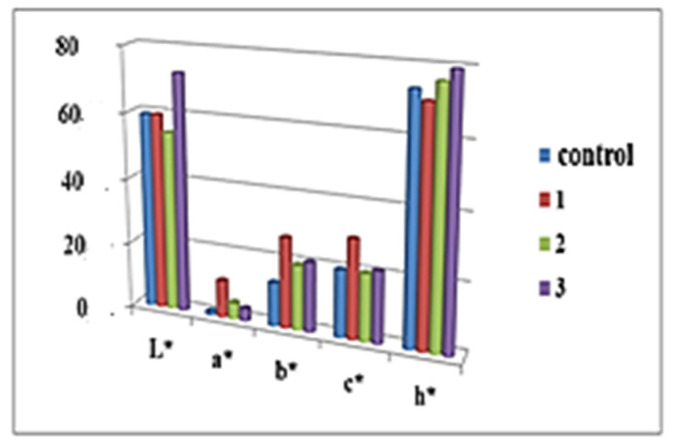
Color indicators of crumb cake elderberry blossom flour (*Sambucus nigra* L.).

**Figure 3 molecules-27-01124-f003:**
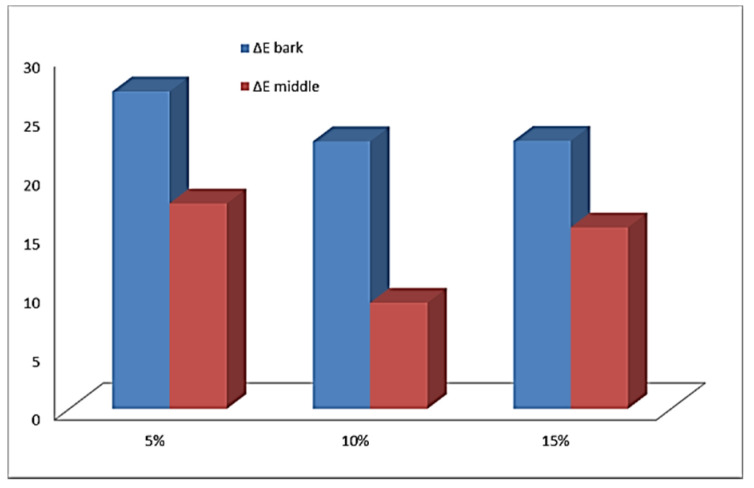
The color difference between Options 1, 2, and 3, and the control for the crust and crumb cake.

**Figure 4 molecules-27-01124-f004:**
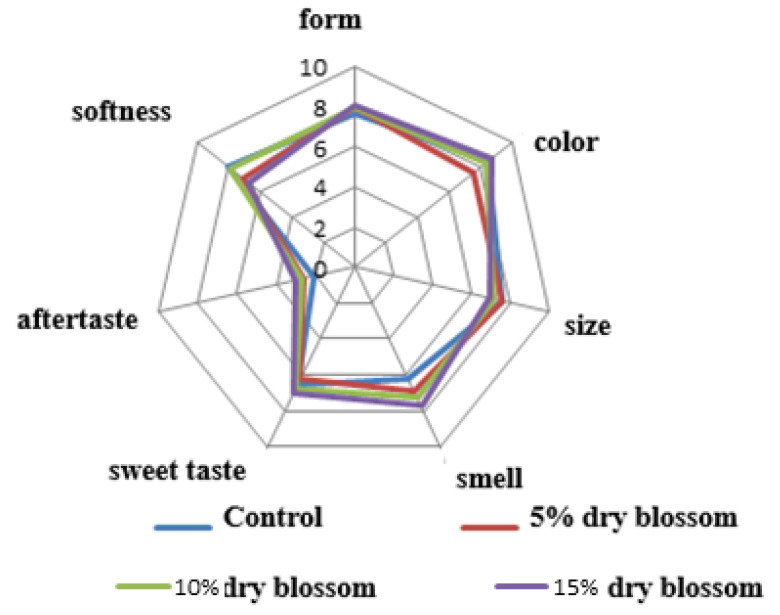
Sensory profile of sponge cakes made with elderberry blossom flour (5, 10, and 15%).

**Figure 5 molecules-27-01124-f005:**
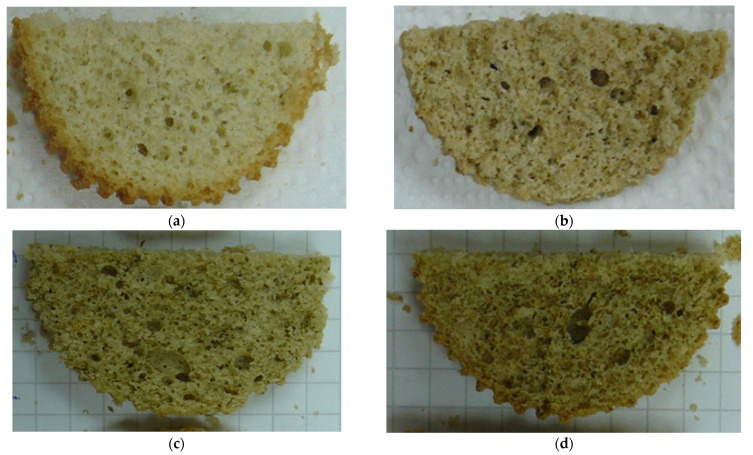
Photo of investigated sponge cakes in the crumb. (**a**) Crumb of Control sample. (**b**) Crumb of Option 1. (**c**) Crumb of Option 2. (**d**) Crumb of Option 3.

**Figure 6 molecules-27-01124-f006:**
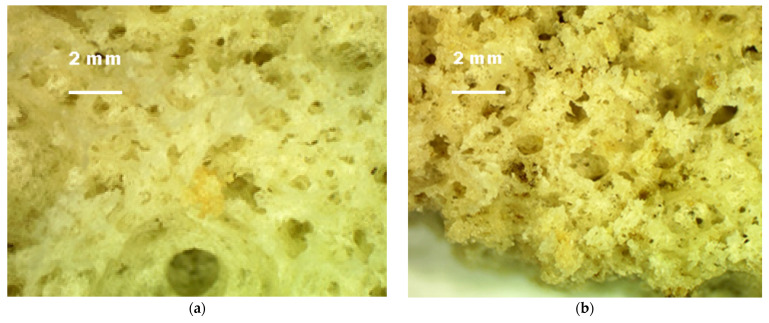
Microscopic images of the Crumb of the sponge cakes. (**a**) Crumb of Control. (**b**) Crumb of Option 1. (**c**) Crumb of Option 2. (**d**) Crumb of Option 3.

**Figure 7 molecules-27-01124-f007:**
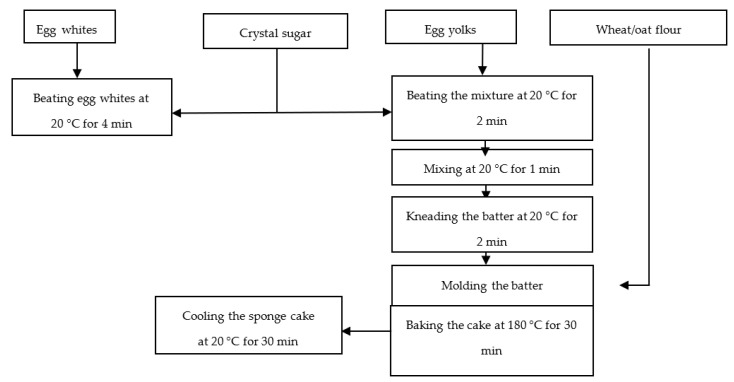
Technological scheme for the preparation of sponge cake control.

**Figure 8 molecules-27-01124-f008:**
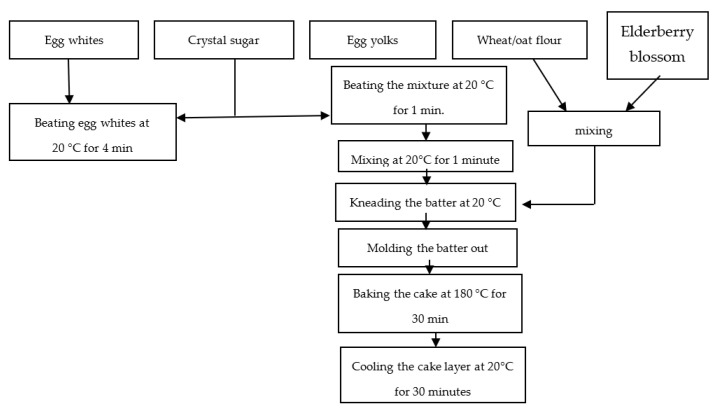
Technological scheme for the preparation of sponge cakes with the addition of *Sambucus nigra* L. flour (5, 10, and 15%).

**Table 1 molecules-27-01124-t001:** Chemical composition of the flours used in the sponge cakes.

	Wheat Flour Type 500	Wholegrain Oat Flour	Dry Elderberry Blossoms /Flour/
Protein, [%]	12.5 ± 0.5	17.3 ± 0.3	2.5 ± 0.05
Lipids, [%]	1.87 ± 0.05	8.7 ± 0.3	-
Carbohydrates, [%]	85.6 ± 0.5	74 ± 0.5	2.55 ± 0.02
Fibers, [%]	3.72 ± 0.02	10.4 ± 0.05	1 ± 0.02
Vit. C, [mg/kg]	-	-	20.1 ± 0.05
Water content, [%]	11.3 ± 0.05	10.6 ± 0.05	4.46 ± 0.04

**Table 2 molecules-27-01124-t002:** Physicochemical characteristics of sponge cakes without and with dried *Sambucus nigra* L. blossom.

Physical Parameters	Type of the Sponge Cake			
	Control	Option 1	Option 2	Option 3
Specific gravity/of the batter/	0.72 ± 0.03	0.81 ± 0.02	0.81 ± 0.02	0.78 ± 0.02
Volume [cm^3^]	207 ± 7	206 ± 5	210 ± 11	208 ± 4
Water absorption capacity [g]	333 ± 8	266 ± 10	263 ± 7	258 ± 10
PH of the finished product	8.00 ± 0.06	7.12 ± 0.09	7.07 ± 0.04	7.18 ± 0.02
Total ash content	0.57 ± 0.04	0.99 ± 0.01	1.57 ± 0.04	1.70 ± 0.03
Moisture, [%]	28.49 ± 0.36	26.01 ± 0.20	27.16 ± 0.17	21.97 ± 0.12

**Table 3 molecules-27-01124-t003:** Textural properties of Control and Options 1, 2, and 3 sponge cakes.

Sample	Hardness, g	Elasticity, %	Brittleness, %
Control	1182 ± 15	71 ± 1	23.5 ± 1
Option 1	1561 ± 20	73.3 ± 3	26 ± 3
Option 2	1640 ± 20	75.7 ± 2	23.6 ± 1
Option 3	2729 ± 25	70.7 ± 5	30.4 ± 7

**Table 4 molecules-27-01124-t004:** Correlation dependencies for sponge cake layers related to textural parameters.

Sample	Correlation Dependencies	$\mathrm{Coefficient}\;\mathrm{of}\;\mathrm{Correlation}\;\left(R^{2}\right)$
Sponge cakes made with elderberry blossom flour	Hardness=93.895C+1071.5	0.917 *
	Elastisity=−229.23 Water content+7714.3	0.95 *
	*d* = 0.024*C* + 0.368	0.93 *

Significance level of * 0.01.

**Table 5 molecules-27-01124-t005:** Physicochemical characteristics of sponge cake without and with dried *Sambucus nigra* L. blossom.

Chemical Properties	Samples			
	Control	Option 1	Option 2	Option 3
Water content, [%]	22.50 ± 0.10	23.75 ± 0.20	25.88 ± 0.60	27.42 ± 0.10
Carbohydrates, [%]	59.60 ± 0.20	55.57 ± 2.40	54.20 ± 1.30	52.84 ± 2.50
Fibers, [%]	2.01 ± 0.20	3.67 ± 0.40	3.57 ± 0.60	3.46 ± 0.30
Protein, [%]	11.58 ± 0.10	12.62 ± 0.20	12.32 ± 0.10	12.02 ± 0.10
Lipids, [%]	6.32 ± 0.03	8.06 ± 0.10	7.89 ± 0.27	7.72 ± 0.11
Energy value, [kcal]	335.00 ± 3.00	352.63 ± 4.00	344.23 ± 5.00	335.85 ± 3.00

**Table 6 molecules-27-01124-t006:** Recipe composition of sponge cake dough, control, and partially substituted wheat flour with 5, 10, and 15% *Sambucus nigra* L. blossom flour and wholegrain oat flour.

	Control	Option 1	Option 2	Option 3
Egg yolk, g	43.23	43.23	43.23	43.23
Egg white, g	96.77	96.77	96.77	96.77
Crystal sugar, g	83.87	83.87	83.87	83.87
Wheat flour type 500, g	100.00	71.61	40.00	-
Wholegrain oat flour, g	-	23.87	50.32	85.16
*Sambucus nigra* L. blossom flour, g	-	4.52	9.68	14.84

## Data Availability

Datasets from the time of this study are available from the respective authors upon reasonable request.

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
