# Peer review of "Incorporation of the Dry Blossom Flour of Sambucus nigra L. in the Production of Sponge Cakes"

_molecules, 2022, doi:10.3390/molecules27031124_

Round 1
Reviewer 1 Report
Paper is good needs some revision

Author Response
Specific points:
In this study, the authors have an interesting idea of using plants rich in biologically active components in order to obtain functional foods.
Introduction - I think that the authors should give an explanation why they decided on wheat and oat flour and why one is whole grain and the other is not.
Classic Bulgarian sponge cake consists of wheat flour, sugar and eggs. Our purpose was to replace wheat flour with a product containing more fibers. Such one is oat flour which well known to the customers.
Tables 2, 3 and 5. Below the table should be explained what control is, as well as options 1, 2 and 3.
Figures 1,2,3,5 and 6 - same comment as above
Their description is given in section 4 Materials and Methods, lines: 235-237
L118-119: „The analyzed wholegrain oat flour and dried blossom flour of Sambucus nigra L. are good sources of biocompounds, therefore suitable in the development of functional foods.” - In my opinion, this is actually a conclusion and I think it should follow the presented results. Namely, I think that this sentence has no place here, at the very beginning of the discussion of the results.
In our opinion that sentence is in correct place. It follows the results and is giving first basic explanation why wholegrain oat flour and dried blossom flour of Sambucus nigra L. are good candidates for use in sponge cakes preparation.
Conclusions - In my opinion, it should be clearly stated which sample showed the best nutritional and functional values.
As described in the text all 3 samples have excellent nutritional and functional values. It is hard to say that one is the best as it depend on the point of view.

Reviewer 2 Report
Specific points:
In this study, the authors have an interesting idea of using plants rich in biologically active components in order to obtain functional foods.
Introduction - I think that the authors should give an explanation why they decided on wheat and oat flour and why one is whole grain and the other is not.
Tables 2,3 and 5. Below the table should be explained what control is, as well as options 1, 2 and 3.
Figures 1,2,3,5 and 6 - same comment as above
L118-119: „The analyzed wholegrain oat flour and dried blossom flour of Sambucus nigra L. are good sources of biocompounds, therefore suitable in the development of functional foods.” - In my opinion, this is actually a conclusion and I think it should follow the presented results. Namely, I think that this sentence has no place here, at the very beginning of the discussion of the results.
Conclusions - In my opinion, it should be clearly stated which sample showed the best nutritional and functional values.
__________________
All my suggestions are for improving the manuscript. I hope all the suggestions are clear.
Best regards.
Author Response
Abstract is corrected.
Recommendation is accepted
The companies of the equipment are added
Determination of TPC and TFC was not part of the current study.
During baking part of ascorbic acid could decompose. Due to the fact that flour is a natural product we are unsure that the amount of ascorbic acid is the same in different batches of flour.
They are presented in the text. The following one is added: we calculate Weight losses after baking according to Hathorn et al. [] For control they are 10.2% while for the samples with Sambucus nigra L weight losses are up to 11.7%.
Physicochemical properties of Sambucus nigra L will be presented in another paper. For pH the following text is added: Probably that is caused by the lower pH of Sambucus nigra L. flour (5.32) than pH of control (5.58).
